# Characterising Eastern Grey Kangaroos (*Macropus giganteus*) as Hosts of *Coxiella burnetii*

**DOI:** 10.3390/microorganisms12071477

**Published:** 2024-07-19

**Authors:** Anita Tolpinrud, Elizabeth Dobson, Catherine A. Herbert, Rachael Gray, John Stenos, Anne-Lise Chaber, Joanne M. Devlin, Mark A. Stevenson

**Affiliations:** 1Asia Pacific Centre for Animal Health, Melbourne Veterinary School, The University of Melbourne, Parkville, VIC 3010, Australiadevlinj@unimelb.edu.au (J.M.D.); 2School of Life and Environmental Sciences, The University of Sydney, Sydney, NSW 2006, Australia; catherine.herbert@sydney.edu.au; 3Sydney School of Veterinary Science, The University of Sydney, Sydney, NSW 2006, Australia; rachael.gray@sydney.edu.au; 4Australian Rickettsial Reference Laboratory, University Hospital Geelong, Geelong, VIC 3220, Australia; john.stenos@barwonhealth.org.au; 5School of Animal and Veterinary Sciences, The University of Adelaide, Roseworthy, SA 5371, Australia; anne-lise.chaber@adelaide.edu.au

**Keywords:** macropods, eastern grey kangaroos, polymerase chain reaction, Q fever, immunofluorescence assay, *Coxiella burnetii*, histopathology, immunohistochemistry, pathology, wildlife

## Abstract

Macropods are often implicated as the main native Australian reservoir hosts of *Coxiella burnetii* (Q fever); however, the maintenance and transmission capacity of these species are poorly understood. The objective of this cross-sectional study was to describe the epidemiology of *C. burnetii* in a high-density population of eastern grey kangaroos (*Macropus giganteus*) in a peri-urban coastal nature reserve in New South Wales, Australia. Blood, faeces and swabs were collected from forty kangaroos as part of a population health assessment. Frozen and formalin-fixed tissues were also collected from 12 kangaroos euthanised on welfare grounds. Specimens were tested for *C. burnetii* using PCR, serology, histopathology and immunohistochemistry. A total of 33/40 kangaroos were seropositive by immunofluorescence assay (estimated true seroprevalence 84%, 95% confidence interval [CI] 69% to 93%), with evidence of rising titres in two animals that had been tested four years earlier. The PCR prevalence was 65% (95% CI 48% to 79%), with positive detection in most sample types. There was no evidence of pathology consistent with *C. burnetii,* and immunohistochemistry of PCR-positive tissues was negative. These findings indicate that kangaroos are competent maintenance hosts of *C. burnetii*, likely forming a significant part of its animal reservoir at the study site.

## 1. Introduction

Animal reservoirs of disease have received significant public and scientific attention in the last few decades, yet the term ‘reservoir’ is often loosely defined. A disease reservoir has most comprehensively been described as one or more epidemiologically connected populations that can permanently maintain a pathogen (termed a ‘maintenance population’), from which infection can be transmitted to a target population of interest [1]. To fulfil this definition, a competent reservoir must be susceptible to infection, capable of onward transmission, be of sufficient size to maintain the pathogen indefinitely without new introductions or sporadic extinctions and be epidemiologically connected to the target host [1,2]. Identifying the reservoir of a multi-host pathogen is inherently difficult and requires a multi-disciplinary approach involving microbiology, pathology, epidemiology, phylogenetic analysis, host biology and ecology, host–pathogen interactions and environmental ecology [2,3]. While the detection of a pathogen, its DNA or antibodies in a potential host population can provide valuable information on where to direct the search for a reservoir, their presence alone cannot be taken as proof of maintenance or transmission capacity [1]. 

*Coxiella burnetii*, the causative agent of Q fever, is a true multi-host pathogen with a near-global distribution and has been identified in over 300 vertebrate species from a wide range of taxa [4]. Q fever is a disease of significant public health importance, and while the majority of human cases are self-limiting, a small percentage of cases experience long-term and potentially fatal sequelae such as endocarditis [5]. The main reservoirs for human infection are domestic ruminants, in which the pathogen is most commonly associated with necrotising placentitis, late-stage abortions and stillbirths [6]. Human infection usually occurs from inhalation of aerosolised, highly environmentally resistant spore-like forms that are shed in large numbers through reproductive products, vaginal secretions, faeces and urine in intensively managed livestock [5,7,8,9]. While sylvatic cycles of infection exist, the wildlife reservoir of *C. burnetii* is relatively poorly understood, particularly as it pertains to the definition provided above [10,11]. 

Macropods (kangaroos and wallabies) are frequently implicated as reservoir hosts of Q fever in Australia, although the majority of the evidence for this assumption is based on scattered prevalence or seroprevalence data [12,13,14,15,16,17]. In most of these studies, sampling was carried out on kangaroos that co-existed with domestic livestock, or the existence of other potential sources of infection was not considered. A small number of risk-factor analyses have linked direct or indirect contact with kangaroos to human cases; however, the links were usually unconfirmed by testing or difficult to differentiate from other potential animal sources [18,19,20,21,22]. One study identified three genotypes of *C. burnetii* in kangaroo pet meat that were identical to strains that have been isolated from Australian human Q fever cases, but because the samples came from packaged commercial pet food at the point of sale, the origin of the animals could not be ascertained [23]. While these findings cumulatively point to a potential macropod reservoir of Q fever, it is largely unknown whether macropods can maintain *C. burnetii* indefinitely without repeated introductions from livestock or whether transmission occurs between or from kangaroos, both crucial determinants of a disease reservoir. 

The objective of this study was to describe the epidemiology of coxiellosis in a peri-urban population of eastern grey kangaroos (*Macropus giganteus*) with a known high seroprevalence of *C. burnetii* antibodies, despite no contact with domestic livestock. Through a combination of molecular detection in tissues and secretions, serology (including serial sampling), histopathology and immunohistochemistry, we aimed to more comprehensively characterise the *C. burnetii* maintenance and transmission potential of kangaroos, as well as identify pathology that might be associated with coxiellosis in macropods. 

## 2. Materials and Methods

### 2.1. Sampling Site and Study Population

This was a cross-sectional study undertaken on a population of kangaroos from the Look At Me Now (LAMN) Headland in Moonee Beach Nature Reserve (30°10′39″ S, 153°11′24″ E) on the north coast of New South Wales (NSW), from November to December 2021. The region has a humid subtropical climate (Cfa) under the Köppen climate classification, with hot wet summers and mild winters [24]. Prevailing winds in the area are predominantly southerly and south-westerly (offshore) in the mornings and north-easterly (onshore) in the afternoons, based on historical weather data recorded at the nearby Coffs Harbour weather station, 15 km south [25]. 

The 24-hectare mostly open, grassy headland is bordered by coastline and suburbia (Appendix A) and is home to a high number of resident free-ranging macropods, including eastern grey kangaroos, red-necked wallabies (*Notamacropus rufogriseus*) and a smaller number of swamp wallabies (*Wallabia bicolor*). The site has one of the highest reported peri-urban kangaroo population densities, with an estimated 5.4 individuals per hectare [26,27]. The kangaroo population has been subject to several repeated population surveys and health assessments. Due to the fragmented habitat and overabundance of kangaroos, the population is of demonstrably poor health, with high levels of parasitism, anaemia and nutritional deficiencies [26,27,28]. Farming in the region is predominantly horticulture, and the kangaroos do not have any known direct contact with domestic livestock. Despite this, a previous retrospective cross-sectional serosurvey on archived kangaroo serum samples collected during health assessments conducted in 2017 and 2018 identified a high seroprevalence to *C. burnetii*, with an estimated true seroprevalence of 81% [29]. 

### 2.2. Sample Collection

Samples for this study were obtained opportunistically as part of a separate independent population health assessment program carried out under permission from the University of Sydney Animal Ethics Committee (University of Sydney AEC 1673). As these were opportunistic samples, the number of animals that were examined was determined in advance and not by a sample size calculation for prevalence estimation. However, based on an expected seroprevalence of 80% [29], test sensitivity and specificity of 97.6% and 98.5% [29], an estimated population of approximately 130 kangaroos [26,27] and an absolute error of 10%, a required sample size to estimate prevalence would be 46 kangaroos [30]. Forty kangaroos were immobilised with zolazepam/tiletamine (Zoletil^®^ 100, Virbac Pty. Ltd., Carros, France) to undergo general health checks, including physical examination, morphological measurements, haematological evaluation and parasite counts. The sedative was delivered intramuscularly with a dart gun (X-Calibre, Pneudart Inc, Williamsport, PA, USA) at a fixed dosage of 125 mg for females and 250 mg for larger males. Once immobilised, the animals were moved to a central processing site for examination and sample collection. Sex, age group, weight, body condition, morphometric measurements, ectoparasite burden and physical examination findings were recorded for each animal. Age group was classified as “adult” or “sub-adult” based on the size of the testes and a minimum body weight of 35 kg (males) or eversion of the teats (females). Blood was collected from the lateral tail vein into ethylenediaminetetraacetic acid (EDTA) and serum gel tubes. Blood in serum gel tubes was allowed to clot before the serum was separated by centrifugation. Faecal samples were collected from each animal where possible for faecal flotation and parasite egg counts, and whole blood, serum and leftover faecal material were frozen for inclusion in this study. Swabs were opportunistically collected from the oral cavity, nasal cavity (nostrils), urogenital sinus and rectum, using sterile, fine-tipped cotton applicator tips that were placed into dry sterile collection tubes. Additionally, swabs were also collected from the pouch of female kangaroos. Approximately 200–300 μL milk was collected from lactating females with young at foot by manually massaging a few drops of milk into microcentrifuge tubes. All samples were frozen and stored at −20 °C until processing. 

Animals deemed to be in reasonable health were recovered and immediately released following examination and sample collection. Kangaroos that have previously been captured at the site have been individually marked with unique colour-coded ear tags and identification numbers to allow for longitudinal monitoring of individual animals, and all first-time captures were assigned a unique ID and tagged before release. 

Kangaroos that showed severe signs of chronic ill health, such as combinations of emaciation, dehydration, hypoproteinaemia, hypoglycaemia, anaemia or permanent debilitating injury, were humanely euthanised on welfare grounds after veterinary assessment. Euthanasia was performed with intravenous pentobarbital, and necropsies were conducted the same day. Sample sets of lungs, heart, liver, mediastinal and/or mesenteric lymph nodes, spleen, kidneys, liver, gastrointestinal tract, bladder, reproductive tract, mammary tissue and skeletal muscle were collected from each animal. Samples were collected as aseptically as possible into individual Ziplock bags and frozen at −20 °C until processing. Paired samples were also collected into 10% neutral buffered formalin for histopathology and immunohistochemistry. Tissues were left to fully fix for 48–72 h before they were transferred to a solution of phosphate-buffered saline (PBS) with 0.1% sodium azide and stored at 4 °C until they could be paraffin embedded.

### 2.3. Serology

Serum samples were tested for antibodies to *C. burnetii* phase I and II antigens using a custom-made, formally validated macropod-specific indirect immunofluorescence assay (IFA), following the protocol described previously [29,31]. Samples were initially screened in duplicate at an initial serum dilution of 1:32, and positive samples were subsequently titrated to determine their endpoint titres, defined as the maximum dilution in which both replicates exhibited strong fluorescence. A known positive and negative control was included on each slide.

### 2.4. Nucleic Acid Extraction

DNA was extracted from approximately 100 mg of faeces using the Isolate II Fecal DNA Kit (Meridian Bioscience Inc., Cincinnati, OH, USA), following the protocol specified by the manufacturer. Tissues, swabs, whole blood, serum and milk were extracted using the Real Genomics HiYield Genomic DNA Mini Kit for blood, bacteria and cultured cells (Real Biotech Corporation, Taipei County, Taiwan). Prior to DNA extraction, swabs were suspended in 600 µL PBS (Vircell S.L., Granada, Spain) and vortexed vigorously for at least 40 s. Extractions were performed on 200 μL of the supernatant, as well as from milk, serum and frozen-thawed whole blood, following the manufacturer’s protocol for frozen whole blood. DNA from tissues was extracted similarly, in duplicate, with a few modifications. Briefly, 15–25 mg of each tissue was manually crushed in 200 μL GT buffer using disposable tissue grinder pestles. Samples were then incubated at 60 °C on a shaker overnight with 10 μL Proteinase K (20 mg/mL; Invitrogen, Thermo Fisher Scientific Inc., Waltham, MA, USA). The remainder of the extraction was then performed following the same protocol as for liquid samples. For each round of extractions, a negative extraction control was included, consisting of 200 μL PBS. An exogenous internal control in the form of a known concentration of suspended *Listeria innocua* culture was added to all samples and extraction controls pre-extraction to determine the extraction efficacy. 

### 2.5. PCR Detection

Real-time polymerase chain reactions were performed to detect the heat shock operon *htpAB*, outer membrane protein *com1* and multicopy insertion sequence IS*1111* genes of the *C. burnetii* genome (Appendix A) [32,33,34]. Assays were performed as described previously on 5 µL DNA template in 25 µL total reactions, using a magnetic induction PCR cycler (Mic) [31]. Extraction success was determined with a PCR targeting a 62 bp DNA fragment from the *lin02483* gene of *L. innocua* on all samples and extraction controls [35]. A negative template control and relevant positive control (*C. burnetii* Nine Mile RSA439 or *L. innocua* culture) were included in all assays. 

Samples with amplification and a cycle threshold (Ct) value of < 40 were considered positive for the target analyte. Due to overall high Ct values exhibited in the positive assays in the initial PCR runs, all samples with a positive assay were retested for all three targets after concentrating the DNA in an attempt to obtain enough genetic material for genotyping. DNA was precipitated by adding 3 μL of 5 M NaCl, 1 μL of 10 mg/mL glycogen (Sigma–Aldrich, Burlington, MA, USA) and 150 μL 100% ethanol to 75 μL of sample. The DNA was then frozen at −20 °C overnight before centrifuging at 20,000× *g* for 10 min. The supernatant was removed, and the pellet was left to air dry before it was resuspended in 15 μL distilled water. The three *Coxiella* PCRs were then repeated on 5 μL each of resuspended DNA extract. Specimens with a positive amplification on any two or more assays were classified as positive, while samples with only one or no amplifications were considered negative for the purpose of the analysis. Tissues were considered positive on an organ-level if at least one of the extracted replicates was PCR-positive, while kangaroos were considered PCR-positive on an animal-level if at least one sample type tested positive. 

### 2.6. Histopathology and Immunohistochemistry

Formalin-fixed tissues were embedded in paraffin, sectioned 4 µm thick and routinely stained with hematoxylin and eosin (Melbourne Histology Platform, the University of Melbourne, Melbourne, VIC, Australia). Stained slides were examined by an anatomic pathologist board-certified by the American College of Veterinary Pathologists. Immunohistochemistry (IHC) and Gimenez staining were performed on slides with a high index of suspicion of *C. burnetii* infection based on (1) positive PCR results across multiple tissues and PCR targets or (2) the presence of appropriate inflammation seen on histopathology, combined with a positive PCR result on that tissue. For the purpose of this study and due to the paucity of information relating to *C. burnetii* infections in macropods, “appropriate inflammation” was regarded as any inflammation with no alternative clear cause, particularly if it was chronic and granulomatous, histiocytic or lymphocytic in nature. IHC was performed using an in-house protocol developed and internally validated previously at the University of Melbourne [36]. Stained sections of a known positive goat placenta with intracytoplasmic *C. burnetii* were used as a positive control.

### 2.7. Statistical Analysis

Fisher’s exact test interpreted at the 5% level of significance was used to test for associations between animal-level prevalence and sex or age group. True seroprevalence was estimated using the Rogan Gladen correction for imperfect tests [37] based on the previously estimated diagnostic test sensitivity (97.6%) and specificity (98.5%) for the IFA [29]. Statistical analyses were performed using the contributed epiR package [38] in R [39].

## 3. Results

### 3.1. Kangaroos Sampled

Forty animals (13 adult and 8 subadult females; 9 adult and 10 subadult males) were captured and sampled as part of this study. Blood, serum and swabs were collected from all 40 kangaroos, while faeces were only available from 35 animals. Milk was obtained from three lactating females with young at foot. 

Twelve animals (six males and six females) were euthanised on welfare grounds and underwent post-mortem examinations and sampling. All the necropsied kangaroos were in poor body condition and had a heavy gastrointestinal and external parasite burden. 

### 3.2. Serology

A total of 33 out of 40 kangaroos (apparent prevalence 82%, 95% CI 67% to 93%) were seropositive to *C. burnetii* on the IFA. Correcting for the use of an imperfect test [37], the estimated true seroprevalence was 84% (95% CI 69% to 93%). The seropositive animals were 14/19 males (8/10 subadults, 6/9 adults) and 19/21 females (8/8 subadults, 11/13 adults), and there was no statistically significant difference between seropositivity and age group or sex (Table 1). Of the seropositive kangaroos, all 33 had antibodies to phase I, while 31 were also seropositive to phase II. The titres for phase I (median 1:4096, range 1:128 to 1:16,384) were consistently higher than those for phase II (median 1:512, range 1:32 to 1:8192).

Five of the adult female kangaroos had also been captured and sampled in 2017 and 2018, and their sera had been tested for antibodies as part of a previous study [29]. Since the previous captures, two of these kangaroos had seroconverted for phase II and showed a significantly increased phase I titre, while the titres for the remaining three kangaroos remained unchanged (Table 2).

### 3.3. PCR Detection of C. burnetii

Of the 40 kangaroos, 26 (65%, 95% CI 48% to 79%) were classed as positive by PCR on at least one sample type. The positive kangaroos comprised 15/19 males (8 subadults, 7 adults) and 11/21 females (6 subadults, 5 adults; Table 1). Overall, 20 kangaroos were positive on both the PCR and the IFA while only one animal (an adult female) was negative on both tests (Table 3). There was no statistically significant association between PCR positivity and age, sex or serological status.

The highest number of specimens that tested positive from one animal was 10 (median = 1, Q1 = 0, Q3 = 2). Eighty-one percent of the PCR-positive animals (n = 21) were positive on at least one ante-mortem specimen (faeces, blood, serum or swabs), while five kangaroos only tested positive on tissue samples collected post-mortem. *Coxiella* DNA was detected in all ante-mortem sample types except serum and milk (Table 4) and in most post-mortem sample types except the testes, uterus and mammary glands (Table 5). Of the 26 positive post-mortem tissue specimens, 9 were positive in duplicate while 17 were only positive on a single replicate. The most frequently positive specimen was faeces, with half of all PCR-positive kangaroos testing positive on this sample type, followed by urogenital swabs (Figure 1). 

For all three targets, Ct values were generally high. The multicopy insertion sequence IS*1111* was most frequently positive (n = 69 specimens) and amplified earlier (Ct range 28.0–38.2) than *htpAB* (n = 21; Ct range 34.3–38.3) and *com1* (n = 24; Ct range 33.9–38.4). Only 16 of the 69 positive specimens amplified all three targets, most of which were swabs and faeces. Concentrating the DNA generally led to a reduction of the Ct value by 1–2 cycles, and in some cases resulted in samples initially negative for *htpAB* or *com1* becoming positive (6 and 15 samples, respectively). A total of 18 samples were amplified only once (for IS*1111*) and were ultimately classified as negative based on the study definition.

### 3.4. Histopathology and Immunohistochemistry

Histopathological examination showed frequent mild to moderate chronic multi-organ inflammation, often associated with the presence of nematodes or protozoal parasites (Appendix A). Immunohistochemistry and Gimenez staining were performed on all tissues from one kangaroo with multiple PCR-positive tissues (animal ID 1121-03, Appendix A), as well as the testicle from one kangaroo with histological signs of epididymitis and PCR detection in epididymal tissue (animal ID 2112-01). Positive immunostaining and Gimenez staining were not identified in any of the evaluated sections.

## 4. Discussion

*Coxiella burnetii* is an organism that infects a diverse range of species, and its epidemiology and reported prevalence vary significantly both between and within species across different geographical areas [14,29,40,41,42,43]. Here, we report the results of a study on an isolated population of kangaroos at LAMN Headland, NSW. In doing so, it should be acknowledged that the inferences made here are locally specific and may not be generalisable to other populations. The results from this study show that these kangaroos have a persistently high prevalence of antibodies to *C. burnetii*, as well as a high apparent animal-level molecular prevalence. There is also evidence of ongoing *C. burnetii* transmission based on the demonstration of infected sub-adults and rising titres in two serially sampled animals. Interestingly, the rate and pattern of PCR detection in tissue samples from necropsied animals and the relative distribution of phase I and II antibodies described in this study were similar to those found in kangaroos co-grazing with domestic ruminants in a Q fever endemic area of Queensland, despite the LAMN population being geographically isolated from livestock [31]. It is important to note that the apparent prevalence estimates presented here must be interpreted with caution due to the small number of necropsied animals and the high proportion of sick animals sampled. 

The mode of transmission of *C. burnetii* between kangaroos is relatively poorly understood. Shedding of large numbers of bacteria during parturition, as is common in placental mammals [5,44], seems unlikely to play a significant role in *C. burnetii* transmission in marsupials due to the absence of a chorioallantoic placenta and the relatively low amount of environmental contamination associated with birth. As such, the most likely routes of excretion based on the above findings appear to be faecal and urogenital, and also possibly in nasal secretions. While the presence of *Coxiella* DNA in kangaroo oral swabs could indicate shedding, it is highly likely these positives represent ingestion of contaminated pasture, considering the high proportion of positive faecal samples, high population density (and associated faecal load) and the limited grazing available at the study site. *Coxiella burnetii* DNA has previously been detected in kangaroo faeces and bladder tissue, although we note that it is unknown whether these findings represent viable bacteria [13,14,31]. However, shedding of viable *C. burnetii* has been demonstrated in faeces, urine and genital swabs in other species, and it is plausible that the same is true for kangaroos [9,45,46,47,48,49]. Molecular detection of *C. burnetii* in oral, nasal, rectal and urogenital swabs has also been described in other species, including wild boar (*Sus scrofa*) and Pacific gulls (*Larus pacificus*) [50,51]. Successful bacterial culture and isolation from these sample types in kangaroos followed by genotyping or DNA sequencing would be valuable next steps but would likely require samples with a higher concentration of *Coxiella* than what was available in this study. 

The high Ct values and patchy distribution of *C. burnetii* in tissues described here are consistent with previous findings in eastern grey kangaroos and other wildlife and indicate that a small amount of genetic material, verging on the limit of detection, was present in the samples [31,52,53]. This also likely explains the discrepancies between the number of samples positive for the highly sensitive multicopy insertion sequence IS*1111* and the single copy genes *htpAB* or *com1*, which both require a higher number of organisms to be present for detection [54]. Although concentrating the DNA resulted in more frequent and earlier amplification, the amount of genetic material in the samples was still considered too low to be suitable for genotyping. This low concentration would explain the absence of positive staining with IHC and Gimenez, which are relatively insensitive compared to molecular detection methods [6]. In light of the concurrent high prevalence of antibodies, it is unknown whether these findings represent low-level, persistent carrier states or repeated infections and clearance. In other species, antibodies have been shown to be less important than cell-mediated immunity in the overall immune response and clearance of *C. burnetii*, and persistent infections in the face of high antibody titres are not uncommon [10,55,56].

Although the number of necropsied kangaroos in this study was small, there was no evidence of pathology associated with PCR-positivity in any of the animals. With an estimated population of approximately 130 kangaroos [26,27], even this relatively small number of animals represents nearly 10% of the total LAMN population. If *C. burnetii* is associated with pathology in these animals, it would be reasonable to expect to find some indication of it in this sample set considering the high prevalence of infection in the population. There is a general paucity of information in the literature on the nature of coxiellosis in marsupials and whether infection is associated with pathology or overt clinical disease in these species. Multifocal splenic necrosis and splenomegaly have been described in a rufous bettong (*Aepyprymnus rufescens*) after experimental infection with *C. burnetii*, while transient pyrexia was recorded in a common brushtail possum (*Trichosurus vulpecula*) and northern brown bandicoots (*Isoodon macrourus*) [57,58]. However, there are no published reports describing clinical signs or pathology in naturally infected marsupials. Although it is possible that focal lesions could have been missed in this study, pathological lesions associated with active proliferation of *C. burnetii* would be expected to contain significantly more genetic material and lower Ct values than those obtained here [44].

The prevalence of exposure to *C. burnetii* in the kangaroos at LAMN Headland is high [29], indicating an unusually high rate of exposure to *C. burnetii* in this location. The high proportion of positive sub-adults, evidence of potential shedding of *C. burnetii*, as well as the rising titres between 2018 and 2021 in two of the five serially sampled kangaroos demonstrate that active transmission is taking place in the population. These findings are significant considering the apparent absence of direct contact with domestic livestock, and they provide strong evidence that the kangaroos could be an important part of the maintenance community for *C. burnetii* at the study site. The public health implications and risk of spillover to humans could be significant and appropriate preventative measures, such as education and vaccination, should be considered for at-risk groups of people in direct or indirect contact with kangaroos.

Although the kangaroo population at LAMN Headland is relatively small and isolated, it is conceivable that the kangaroos maintain *C. burnetii* in a hyperendemic state. The bacterium is highly infectious, environmentally stable and is known to be capable of causing persistent infections in a range of species [55,59,60,61]. Habitat fragmentation leading to higher host population densities (e.g., due to reduced dispersal opportunities) can drive an increase in density-dependent pathogen transmission [62,63,64]. The combination of an infectious pathogen with a low host mortality and morbidity rate, poor immune clearance, high host population density and heavy bacterial contamination of grazing land could be contributing factors to the high prevalence. Whether other wildlife species in the area, such as birds, wallabies, rodents or other small mammals, are also involved in the maintenance community and transmission cycle remains unknown and warrants investigation. However, the social behaviour and gregariousness of eastern grey kangaroos, coupled with the high kangaroo density at LAMN Headland likely provide ideal opportunities for *C. burnetii* transmission and maintenance, possibly enhanced by immune compromise due to the overall poor population health and nutritional status [26,64]. Longitudinal monitoring with serial testing of the kangaroos, facilitated by individual animals being easily identifiable by their ear tags, would provide valuable insight into the transmission dynamics and epidemiology of *C. burnetii* in this population. Although less likely, it is also feasible that wind-borne dissemination of *C. burnetii* from infected livestock could act as a source of repeated introductions into the kangaroo population. While there are no known livestock farms near the headland, there are cattle farms several kilometres further inland, upwind from the prevailing morning winds. Windborne spread from infected farms has been implicated as a cause of several outbreaks of Q fever in humans [65,66,67,68,69]. Prevalence surveys of domestic ruminants, in addition to investigation of the human incidence of Q fever in the wider Coffs Harbor area, would help provide a more complete understanding of the Q fever epidemiology in the region.

## 5. Conclusions

The results of this study show that the eastern grey kangaroo population at Look At Me Now Headland has a high serological and molecular prevalence of *C. burnetii*, with evidence of active, sustained transmission and possible shedding. Although kangaroos seem to develop disseminated systemic coxiellosis, infection does not appear to be associated with overt pathology. In the absence of direct contact with the traditional livestock reservoir, this may indicate that eastern grey kangaroos are competent maintenance hosts for *C. burnetii* and potentially form a significant part of the reservoir community in this location, likely exacerbated by the increased urbanisation that has contributed to the fragmented habitat and high kangaroo density at the site. It should be noted that these inferences are specific to this particular study site and cannot necessarily be generalised to other kangaroo populations. Although it is currently unknown whether the bacteria detected in the excretions and secretions of the kangaroos represent viable bacteria, careful consideration should be given to the possible public health implications of the spillover of Q fever to humans who live, work or recreate in the area. Similarly, incorporating sustainable wildlife management practices into urban planning, such as wildlife corridors and preservation of suitable habitats, could be beneficial to limit the risk of zoonotic disease spillover to humans and domestic animals.

## Figures and Tables

**Figure 1 microorganisms-12-01477-f001:**
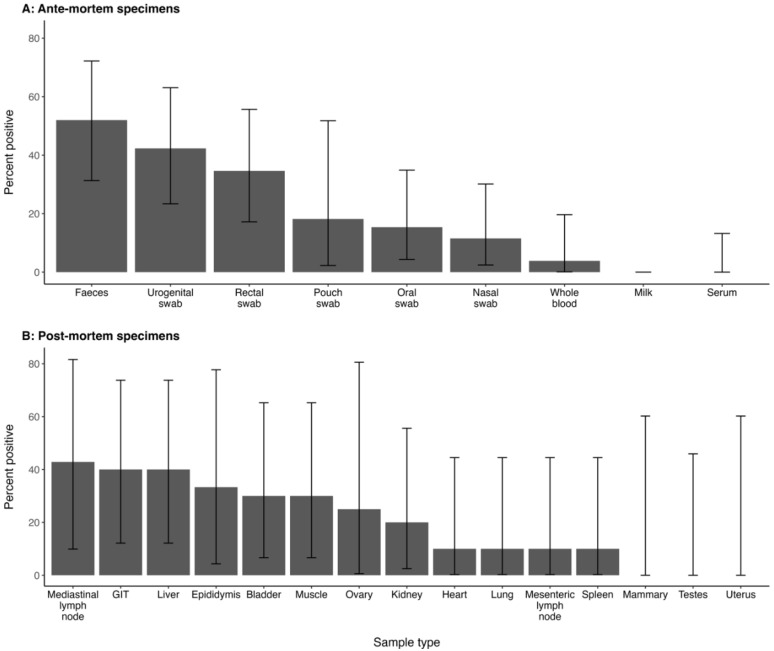
Ranked bar plot showing the percentage likelihood of a sample type being positive for *C. burnetii* in PCR-positive eastern grey kangaroos (n ≤ 26) sampled at Look At Me Now Headland, NSW. (**A**) ante-mortem specimens and (**B**) post-mortem specimens (sample size shown in Table 5). Error bars represent the 95% confidence intervals. GIT = gastrointestinal tract.

**Table 1 microorganisms-12-01477-t001:** Breakdown of the apparent prevalence and the 95% confidence intervals (CI) for the PCR and IFA between sex and age groups.

Strata	Total Tested	PCR		IFA	
		n pos	% pos (95% CI)	n pos	% pos (95% CI)
**Sex:**					
Male	19	15	79 (54, 94)	14	74 (49, 91)
Female	21	11	52 (30, 74)	19	90 (70, 99)
**Age group:**					
Adult	22	12	55 (32, 76)	17	77 (55, 92)
Subadult	18	14	78 (52, 94)	16	89 (65, 99)
*Total*	*40*	*26*	*65 (48, 79)*	*33*	*82 (67, 93)*

**Table 2 microorganisms-12-01477-t002:** Phase I and II antibody titres in serially sampled eastern grey kangaroos at Look At Me Now Headland, NSW, in 2017/2018 and 2021.

	2017/2018		2021		
Kangaroo ID	Phase I	Phase II	Phase I	Phase II	Trend
306	Negative	Negative	Negative	Negative	No change
307	Negative	Negative	Negative	Negative	No change
309	1:2048	Negative	1:4096	1:512	Increasing
323	1:32	Negative	1:1024	1:256	Increasing
341	1:16,384	1:8192	1:16,384	1:8192	No change

**Table 3 microorganisms-12-01477-t003:** Contingency table with the animal-level PCR and IFA results for the eastern grey kangaroos sampled at Look At Me Now Headland, NSW.

	IFA +	IFA −	Total
**PCR +**	20	6	26
**PCR −**	13	1	14
**Total**	33	7	40

**Table 4 microorganisms-12-01477-t004:** Number of eastern grey kangaroos sampled at Look At Me Now Headland, NSW, classed as positive for *C. burnetii* by PCR on ante-mortem samples.

Sample Type	Total Tested	n pos	% pos (95% CI)
Whole blood	40	1	2.5 (0.1, 13)
Serum	40	0	0 (0, 8.8)
Nasal swab	40	3	7.5 (1.6, 20)
Oral swab	40	4	10 (2.8, 24)
Urogenital swab	40	11	28 (15, 44)
Rectal swab	40	9	23 (11, 39)
Faeces	35	13	37 (22, 55)
Pouch swab	21	2	9.5 (1.2, 30)
Milk	3	0	0 (0, 71)
*Total kangaroos*	*40*	*21* *	*52 (36, 68)*

* with one or more positive samples.

**Table 5 microorganisms-12-01477-t005:** Number of eastern grey kangaroos sampled at Look At Me Now Headland classed as positive for *C. burnetii* by PCR on tissues collected post-mortem.

Tissue—Sample Type	Total Tested	n pos	% pos (95% CI)
Heart	12	1	8.3 (0.2, 38)
Lung	12	1	8.3 (0.2, 38)
Mediastinal lymph node	7	3	43 (10, 82)
Mesenteric lymph node	12	1	8.3 (0.2, 38)
Spleen	12	1	8.3 (0.2, 38)
Liver	12	4	33 (10, 65)
Kidneys	12	2	17 (2.1, 48)
Gastrointestinal tract	12	4	33 (10, 65)
Skeletal muscle	12	3	25 (5.5, 57)
Bladder	12	3	25 (5.5, 57)
Mammary gland	6	0	0 (0, 46)
Ovaries	6	1	17 (0.4, 64)
Uterus	6	0	0 (0, 46)
Testes	6	0	0 (0, 46)
Epididymis	6	2	33 (4.3, 78)
*Total kangaroos*	*12*	*10* *	*83 (52, 98)*

* with one or more positive samples.

## Data Availability

The original data presented in the study are openly available in Open Science Framework at https://osf.io/b2wp4 or DOI: 10.17605/OSF.IO/B2WP4.

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
