# Peer review of "Characterising Eastern Grey Kangaroos (Macropus giganteus) as Hosts of Coxiella burnetii"

_microorganisms, 2024, doi:10.3390/microorganisms12071477_

Round 1

Reviewer 1 Report

Comments and Suggestions for Authors

The work written by Tolpinrud et al. describes evidence of exposure to Coxiella burnetii and molecular positivity in several matrices derived from gray kangaroos. The work is very interesting as it describes some characteristics of Coxiella burnetii infection in the kangaroo, considered an occasional host/reservoir/sentinel of this pathogen. The sections are well structured, the methods used are adequate, and the conclusions are correct and balanced. Some minor comments below:
1) Abstract 18: Here it would seem that the samples are paired when, in fact, they are not.
2) Abstract 22–23: no CI is needed in the abstract.
3) Introduction (but also discussion) Line 60: I would insert information about other occasional hosts of Coxiella, for example, dogs, pigs (recent evidence has been found in the Campania region for both species, Brazil for dog, Korea for pigs), etc., and I would underline the differences and similarities of the trend of C. burnetii infection in these species.
4) 115: The sampling was for convenience and did not derive from statistical formulas. It should be specified.
5) Line 175: A table with the primers and protocols used would be helpful.
6) Line 240: Since it is an epimeiolgoic study, I expected a summary table of molecular and serological positivities as the first table (perhaps with subdivision by sex, etc.). I ask the authors to re-evaluate the order and content of the tables presented.
7) The discussions are poor in comparing the prevalences obtained with animals of similar or different species.

8) Authors advise considering an image for illustrative purposes of an IFA (e.g., with positive and negative.

Author Response

1) Abstract 18: Here it would seem that the samples are paired when, in fact, they are not.

Response 1: We have reviewed the text carefully and as there is no mention of paired samples in the abstract, we are unsure what the reviewer is referring to. For this reason, we have left the text as is. If the editor/reviewer can clarify the issue, we would be happy to revise as necessary.

2) Abstract 22–23: no CI is needed in the abstract.

Response 2: We have included confidence intervals as they provide the level of uncertainty around the estimates. We regard this as particularly important here considering the relatively small sample size and by providing this in the abstract, it hopefully prevents inappropriate inferences from being made due to interpretation of point estimates alone, by readers who only read the abstract.

3) Introduction (but also discussion) Line 60: I would insert information about other occasional hosts of Coxiella, for example, dogs, pigs (recent evidence has been found in the Campania region for both species, Brazil for dog, Korea for pigs), etc., and I would underline the differences and similarities of the trend of C. burnetii infection in these species.

Response 3: The subject of this paper provides insight into the epidemiology of a very specific population of kangaroos, and for this reason, we are not of the belief that discussing Q fever in other species in other geographical areas will be helpful in introducing the aims of our study, particularly because the reported prevalences are so location/context specific and highly variable within and between species. To acknowledge this and make this point clearer to the reader, we have added the following statement to the beginning of the discussion, lines 305 - 310 in the revised manuscript:

Coxiella burnetii is an organism that infects a diverse range of species, and its epidemiology and reported prevalence vary significantly both between and within species across different geographical areas [14,30,40-43]. Here, we report the results of a study on an isolated population of kangaroos at LAMN Headland, NSW. In doing so, it should be acknowledged that the inferences made here are locally specific and may not be generalisable to other populations.”

4) 115: The sampling was for convenience and did not derive from statistical formulas. It should be specified.

Response 4: Thank you for this comment. This is a great point and the text has been amended to reflect this, lines 109 - 114 in the revised manuscript: “As these were opportunistic samples, the number of animals that were examined was determined in advance and not by a sample size calculation for prevalence estimation. However, based on an expected seroprevalence of 80% [29], test sensitivity and specificity of 97.6% and 98.5% [29], an estimated population of approximately 130 kangaroos [26,27] and an absolute error of 10%, a required sample size to estimate prevalence would be 46 kangaroos [30].”

5) Line 175: A table with the primers and protocols used would be helpful.

Response 5: Thank you for this suggestion; a supplementary table (S1) has been provided.

6) Line 240: Since it is an epimeiolgoic study, I expected a summary table of molecular and serological positivities as the first table (perhaps with subdivision by sex, etc.). I ask the authors to re-evaluate the order and content of the tables presented.

Response 6: Thank you for this comment. We have added a new table (Table 1) on page 6 with this information.

7) The discussions are poor in comparing the prevalences obtained with animals of similar or different species.

Response 7: This comment appears to be related to comment 3 - please refer to our response above.

8) Authors advise considering an image for illustrative purposes of an IFA (e.g., with positive and negative.

Response 8: Although we do agree that this is a good suggestion, we unfortunately do not have any images of sufficient quality available from the IFA testing.

Reviewer 2 Report

Comments and Suggestions for Authors

The overall goal of the here presented study was to clarify whether grey kangaroos represent a reservoir for Cb. A population (n=40) of grey kangaroos with known high seroprevalence (3-4 years before this study) for Q fever and w/o contact to domestic ruminants was analyzed. The applied methodology is appropriate and sound. The animals were of general bad health conditions. Serology and PCR analysis of tissues indicated an ongoing low-level infection or continuous contact with Cb. High positivity of faeces and urogenital swab samples imply that faeces/urine may be the main shedding route and cause environmental contamination. Overall negative pathology and IHC findings are likely due to the low bacterial load in organs. The authors discussed that the overall bad health status of the animals may have an influence on the results and that PCR detection is no proof of viable bacteria.

Concluding from the here described finings, that kangaroos may play a significant role as reservoir or transmission is not appropriate. The examined population is in bad health condition which may favor a persistent or repeated Cb infection and does not reflect the true status. Further the area is highly populated with other likely susceptible species.

Line 55: Cb does not produce spores. The morphological form is termed small cell variant (SCV) which is supposed to be more resistant to environmental stress. But there are no publications proving that only SCVs are resistant. All publications use a mixture of SCV and LCVs.

Line 56-57: the bacteria are shed in high numbers in afterbirth material, but less in urine and faeces independently from the livestock system.

Line 125: What do you mean by “(specific sites?)”. Please clarify.

Line 160: please specifiy the PBS used. Self made and recipe/commercial and manufacturer/pH?

Line 186: concentration of glycogen

Line 188: provide x g not rpm; centrifugation was carried out for 1 min only? This is unusual.

Author Response

Comment 1: Concluding from the here described findings, that kangaroos may play a significant role as reservoir or transmission is not appropriate. The examined population is in bad health condition which may favor a persistent or repeated Cb infection and does not reflect the true status. Further the area is highly populated with other likely susceptible species.

Response 1: Thank you for this comment. We would like to stress that our conclusion that kangaroos may play a significant role as maintenance hosts relates to this specific study site, which we argue is supported by the findings summarised on lines 310 - 314, 373 - 380 and 411 - 413 in the revised manuscript. We have put forward a number of hypothesised factors we believe may have influenced these findings (lines 329 - 330, 384 - 391, 395 - 398, and 415 - 419). To make it clearer that our conclusion relates to this specific population, we have added the following sentence to lines 419 - 421 in the conclusion: “It should be noted that these inferences are specific to this particular study site and cannot necessarily be generalised to other kangaroo populations.”

We have also added a similar clarification to the beginning of the discussion (lines 305 - 310): “Coxiella burnetii is an organism that infects a diverse range of species, and its epidemiology and reported prevalence vary significantly both between and within species across different geographical areas [14,30,40-43]. Here, we report the results of a study on an isolated population of kangaroos at LAMN Headland, NSW. In doing so, it should be acknowledged that the inferences made here are locally specific and may not be generalisable to other populations.”

While the kangaroo population is the most plausible maintenance population at the study site for the reasons stated, we agree that there is a need to look at other species in the area, as is pointed out in the manuscript (lines 392 - 394).

Although the population is indeed in poor general health, we point out that the molecular prevalence in the necropsied kangaroos described here is similar to that of another population tested in Queensland, which was in good health the time of sampling (Tolpinrud A, Tadepalli M, Stenos J, Lignereux L, Chaber A-L, Devlin JM, et al. (2024) Tissue distribution of Coxiella burnetii and antibody responses in macropods co-grazing with livestock in Queensland, Australia. PLoS ONE 19(5): e0303877. https://doi.org/10.1371/journal.pone.0303877).

Comment 2: Line 55: Cb does not produce spores. The morphological form is termed small cell variant (SCV) which is supposed to be more resistant to environmental stress. But there are no publications proving that only SCVs are resistant. All publications use a mixture of SCV and LCVs.

Response 2: We have used the term spore as it has been used regularly in the historical literature (some examples are included below). We acknowledge that as the microbiology field improves, the term ‘spore’ may or may not still be appropriate and for this reason, we have edited the text to refer to ‘spore-like forms’ instead.

  • Williams, J. C. and Thompson, H. A. 1991. Q fever: the biology of Coxiella Burnetii. CRC Press.
  • McCaul TF, Dare AJ, Gannon JP, Galbraith AJ. In vivo endogenous spore formation by Coxiella burnetii in Q fever endocarditis. J Clin Pathol. 1994 Nov;47(11):978-81. doi: 10.1136/jcp.47.11.978.
  • Oswald, W. and Thiele, D. (1993), A Sporulation Gene in Coxiella burnetii?. Journal of Veterinary Medicine, Series B, 40: 366-370. Doi: 10.1111/j.1439-0450.1993.tb00151.x
  • McCaul TF, Williams JC. 1981. Developmental cycle of Coxiella burnetii: structure and morphogenesis of vegetative and sporogenic differentiations. J Bacteriol 147: doi: 10.1128/jb.147.3.1063-1076.1981
  • Eldin C, Mélenotte C, Mediannikov O, Ghigo E, Million M, Edouard S, Mege JL, Maurin M, Raoult D. From Q Fever to Coxiella burnetii Infection: a Paradigm Change. Clin Microbiol Rev. 2017 Jan;30(1):115-190. doi: 10.1128/CMR.00045-16.

Comment 3: Line 56-57: the bacteria are shed in high numbers in afterbirth material, but less in urine and faeces independently from the livestock system.

Response 3: We are unsure if the reviewer is requesting a change or making a comment and have therefore left the text as is. While the amount of bacteria in other sample types may not be as numerous as in birth products, we have provided several references to support that C. burnetii is shed in significant numbers in faeces.

Comment 4: Line 125: What do you mean by “(specific sites?)”. Please clarify.

Response 4: This was part of a previous communication between co-authors that had been missed in the editing process and has now been removed. Thank you for pointing out this mistake.

Comment 5: Line 160: please specify the PBS used. Self made and recipe/commercial and manufacturer/pH?

Response 5: The PBS is commercial and the manufacturer has been added on line 166 in the revised manuscript.

Comment 6: Line 186: concentration of glycogen

Response 6: Thank you for this – we have added 10 mg/ml to line 191 of the revised manuscript.

Comment 7: Line 188: provide x g not rpm; centrifugation was carried out for 1 min only? This is unusual.

Response 7: Thank you for bringing this to our attention – the text was supposed to say 10 min and has been corrected. The centrifugation force has been converted to 20,000 x g.